# Circadian Gene *cry* Controls Tumorigenesis through Modulation of Myc Accumulation in Glioblastoma Cells

**DOI:** 10.3390/ijms23042043

**Published:** 2022-02-12

**Authors:** Patricia Jarabo, Carmen de Pablo, Amanda González-Blanco, Sergio Casas-Tintó

**Affiliations:** Instituto Cajal, CSIC, 28002 Madrid, Spain; cdpcarreras@gmail.com (C.d.P.); amanda11_4@hotmail.com (A.G.-B.)

**Keywords:** cancer, neurodegeneration, glioma, *Drosophila*, disease model, PI3K, EGFR, genetics

## Abstract

Glioblastoma (GB) is the most frequent malignant brain tumor among adults and currently there is no effective treatment. This aggressive tumor grows fast and spreads through the brain causing death in 15 months. GB cells display a high mutation rate and generate a heterogeneous population of tumoral cells that are genetically distinct. Thus, the contribution of genes and signaling pathways relevant for GB progression is of great relevance. We used a *Drosophila* model of GB that reproduces the features of human GB and describe the upregulation of the circadian gene *cry* in GB patients and in a *Drosophila* GB model. We studied the contribution of *cry* to the expansion of GB cells and the neurodegeneration and premature death caused by GB, and we determined that *cry* is required for GB progression. Moreover, we determined that the PI3K pathway regulates *cry* expression in GB cells, and in turn, *cry* is necessary and sufficient to promote Myc accumulation in GB. These results contribute to understanding the mechanisms underlying GB malignancy and lethality, and describe a novel role of Cry in GB cells.

## 1. Introduction

Glioblastoma (GB) is the most common and aggressive type of glioma among all brain tumors, and it accounts for 57.3% of all gliomas [1]. It was classified in 2016 as a WHO grade IV diffuse oligodendroglial and astrocytic brain tumor, but the most recent classification (2021) includes these type of tumors in the “Gliomas, glioneuronal tumors, and neuronal tumors” group, termed as Glioblastoma, IDH-wildtype [2]. Despite current treatments, the median survival of GB patients is 15 months [3], and it is estimated that only 6.8% of patients survive five years after diagnosis [1]. To understand the genetic, molecular and cellular bases of gliomagenesis is fundamental for the development of effective therapies. In terms of histopathology and genetic expression, GB is a very heterogeneous type of tumor, even within the same patient [4]. However, there are common mutations in GB affecting different pathways that show mutual exclusion: the p53 pathway, the Rb pathway and components of the PI3K pathway [5].

Previous studies from our lab used a GB model in *Drosophila*, developed by Read and collaborators in 2009, that recapitulates key aspects of the disease both genetically and phenotypically [6,7,8,9,10,11,12,13]. This model is based on the expression of constitutively active forms of the *epidermal growth factor receptor* (*EGFR^λ^*) and *phosphatidyl inositol 3 kinase* (*PI3K*) catalytic subunit (*dp110^CAAX^*) (orthologues of EGFR and PI3K catalytic subunit in *Drosophila*, respectively). We used the binary expression system Gal4/UAS [14] to express *EGFR^λ^* and *PI3K dp110^CAAX^* specifically in glial cells under the control of *repo-Gal4* driver [6]. The co-activation of EGFR and PI3K signaling pathways in *Drosophila* glial cells reproduces the cascade of signaling events that occurs in GB patients [6]. In consequence, GB cells upregulate *myc* expression, which is essential for tumoral transformation, and the glial tumor cell numbers increase along with the expansion of the glial membrane. As a result, GB progression causes a reduction in the number of synapses in neighboring neurons and premature death [6,9,15]. Furthermore, EGFR and PI3K pathway co-activation regulates processes such as progression and entry into the cell cycle and protein synthesis [6,7].

*c-myc* is one of the oncogenes most amplified in human cancer, including GB. About 60%–80% of human GB cases show elevated Myc levels [16]. Myc regulates cell proliferation, transcription, differentiation, apoptosis and cell migration. It is the point where EGFR and PI3K pathways converge; thus, Myc is considered essential for GB transformation [6,16,17,18]. Furthermore, in vitro and in vivo studies have shown that *myc* inhibition prevents glioma formation, inhibits cell proliferation and survival and even induces disease regression [16,19]. These features are conserved in *Drosophila* [6].

In the recent years, the study of alterations in circadian rhythm genes has emerged in different types of cancer, including GB [20]. Previous reports suggested that circadian rhythm genes play essential roles in different aspects of tumor progression. The central clock organizes the oscillations and rhythmicity of the physiological processes and modulates the expression of genes related to cell proliferation or differentiation, such as cell cycle components [21], proto-oncogenes and tumor suppressors [22].

In mammals, the structure responsible for coordinating circadian behavior throughout the body is the suprachiasmatic nucleus (SCN), located in the anterior region of the hypothalamus and made up of about 50,000 neurons in humans [23]. All the neurons that compose the central clock express the core circadian genes that control the oscillations that organize the cycles of the whole organism in absence of environmental cues. Furthermore, synchronization of the internal clock with light/dark cycles relies on cryptochrome protein (Cry), a blue light photopigment expressed in certain subsets of clock neurons. Cry is a receptor of near-UV/blue light and a regulator of gene expression that belongs to the group of DNA photolyases. It was suggested that the last universal common ancestor (LUCA) had one or several photolyases, supporting the evolutionary conservation of cryptochrome genes [24]. However, the mammalian gene that plays the role of *Drosophila cry* remains unknown. Interestingly, *Drosophila* Cry also acts as the mammalian Cry when expressed in peripheral clocks [25]. Besides, *cry1* expression is androgen responsive, Cry1 regulates DNA repair and the G2/M transition and it is associated with poor outcome in prostate cancer and colorectal cancer.

Regarding GB, studies in patients with primary gliomas show an association between a specific *per1* variant with overall glioma risk. Several circadian genes, including *cry1*, exhibited differential expression in GB samples compared to control brains as described in the literature [26,27], and in human cancer gene expression databases (https://www.proteinatlas.org, accessed on 1 February 2022; https://cancer.sanger.ac.uk/, accessed on 10 January 2022). Besides, the expression of the circadian gene *clk* is significantly enhanced in high-grade gliomas and correlates with tumor progression [28]. Moreover, *per1* and *per2* expression increases the efficacy of radiotherapy also in GB cells [29].

Furthermore, high levels of *cry1* inversely correlate with median survival in GB patients, acting as signal of poor prognosis (http://gepia.cancer-pku.cn/detail.php?gene=CRY1, accessed on 1 February 2022). Still, the functional mechanism of Cry in cancer susceptibility and carcinogenesis remains unsolved.

Different studies show a relationship between Cry and Myc [30]; c-Myc levels decrease in *cry1*/*cry2* null mutant mice [31]. Besides, *cry1* expression is induced by Myc in GB cells in culture [32].

Taking into account the deregulation in the expression of circadian genes in tumor tissues and the pre-established relationship between Cry and *myc*, which is a key player in GB, here we show that *cry* is regulated by PI3K pathway, *cry* expression enhances Myc accumulation in GB cells and it is necessary for GB progression.

## 2. Results

### 2.1. Cry Expression in Glioblastoma

To determine if *cry* expression was affected in glioma samples, we extracted RNA from the heads of 7-day-old adult control and glioma flies. Quantitative RT-PCR results (see Table 1 in Materials and Methods) indicate that *cry* mRNA levels are 50 times higher in glioma samples as compared to controls (Figure 1A). This result goes in line with the data retrieved from TCGA-GBM dataset (at http://gliovis.bioinfo.cnio.es/, accessed on 1 February 2022) that indicate a significant increase of *cry1* mRNA levels (RNA-seq) in GB samples, as compared to non-tumor tissue.

Next, to determine if *cry* upregulation occurs in GB cells, we used a specific reporter line that generates a green fluorescent protein (GFP) tagged form of Cry (GFP-Cry) and visualized adult brains in confocal microscopy. The images show the GFP signal (Cry) and glial membrane marked in red with myristoylated red fluorescent protein (mRFP) (Figure 1B–E,B’–E’). The quantification of GFP-Cry and mRFP co-localization is higher in glioma samples than in controls (Figure 1B,C,F) suggesting an accumulation of Cry in glioma cells. This signal is restored to control levels upon *cry* knockdown by means of RNAi expression in glial or glioma cells (Figure 1D–F).

Next, we analyzed human mRNA expression databases for Glioblastoma multiforme (http://gliovis.bioinfo.cnio.es/, accessed on 10 January 2022). The results indicate that *cry* in GB patients is transcriptionally upregulated in primary tumors (Figure 1G) and *cry1* upregulation correlates with worse prognosis (Figure 1H). Moreover, *cry1* is also upregulated in secondary GB (Figure 1I) and correlates with poor prognosis in secondary GB patients (Figure 1J). All together, these results indicate that *cry* is transcriptionally upregulated in GB cells in *Drosophila* and patients and suggest a role in GB malignancy and aggressiveness.

### 2.2. Cry Mediates GB Progression and Neurodegeneration

To determine the contribution of *cry* to GB progression, we used a previously validated protocol to quantify tumor progression and the associated neurodegeneration in *Drosophila* [7,9,11]. We stained adult control brains and compared them with GB, GB + *cryRNAi* and wt brains expressing *cryRNAi* in glial cells. We used a specific antibody against repo to visualize the nuclei of all glial cells and quantified the number of glial cells in the confocal images (Figure 2A–E). The results indicate that GB samples have a significant increase in the number of glial cells compared to control samples, but this increase depends on *cry* expression (Figure 2A–C,E). Besides, knockdown of *cry* in normal glia does not alter the number of glial cells (Figure 2D,E). In addition, we quantified the volume of glial membrane. We used Imaris software to measure the volume of the red signal that corresponds to a myristoilated form of RFP (mRFP) expressed in glial cells under the control of *repo-Gal4*. The quantification of the volume show a significant expansion of glial membrane in GB compared to control samples, but this increase depends on *cry* expression (Figure 2A’–C’,F). Again, knockdown of *cry* in normal glia does not alter the volume of glial membrane (Figure 2D’,F). These results suggest that *cry* expression is required for GB progression, but not for normal glia development.

Next, we studied the impact of GB progression and *cry* expression in neighboring neurons. We counted the number of synapses in motor neurons of adult neuromuscular junction (NMJ), a standardized tissue to study neurodegeneration [9,11,33]. To visualize synapses, we used an anti-Brp antibody (nc82) to detect active zones in the neurons, and counted the number of synapses in control samples, GB, GB + *cryRNAi* and normal glia + *cryRNAi* (Figure 2G–J). The quantification of synapse number (Figure 2K) shows that GB induction provokes a significant reduction in the number of synapses as compared to control samples, compatible with a neurodegenerative process. This effect was previously described [7,9,11] as a consequence of GB progression. Moreover, *cry* knockdown in GB prevents the reduction in the number of synapses, and *cryRNAi* expression in normal glial cells does not cause any detectable change in the number of synapses. Finally, we aimed to determine the systemic effect of *cry*. We expressed *cryRNAi* in glia or GB cells, and we analyzed the life span of adult flies. The results show that GB causes a significant reduction of life span and a premature death, which is prevented by *cryRNAi* expression in GB cells. Moreover, *cryRNAi* expression in normal glial cells does not reduce lifespan but causes a significant increase in the average lifespan (Figure 2L).

### 2.3. Signaling Pathway to Control Cry Upregulation

To decipher the specific signaling pathway responsible for *cry* transcriptional activation in GB cells, we analyzed the contribution of the two main pathways activated in this model of GB, EGFR and PI3K. Both pathways converge in the expression of the gene *myc* (see Figure 3A for detailed genetic epistasis in GB). Thus, we analyzed the contribution of *PI3K*, *EGFR* and *myc* to *cry* upregulation. We measured the fluorescent signal of GFP-*cry* reporter in control adult brains (Figure 3B–B″) and compared it with adult brains upon expression of the constitutively active forms of *PI3K* (Figure 3C–C″) or *EGFR* (Figure 3D–D″) in glial cells (under the control of *repo-Gal4*). In addition, we analyzed the GFP-cry signal in glial cells upon *myc* upregulation (Figure 3E–E″). We quantified in the confocal images the signal of GFP that co-localizes with glial membranes (mRFP) (Figure 3F). The results indicate that *PI3K* expression is sufficient to increase *GFP-cry* signal but not *EGFR* or *myc* overexpression. These results suggest that *PI3K* upregulation induces *cry* transcription, and *EGFR* or *myc* expression do not induce *cry* expression in glial cells.

### 2.4. Cry Regulates Myc Expression in Glial Cells

Next, to determine the epistatic relation between *cry* and *myc*, we analyzed Myc protein accumulation in glial cells upon *cry* expression. First, to analyze if Cry is sufficient to cause an increase in Myc protein levels, we used a specific antibody against Myc and analyzed Myc signal levels upon *cry* overexpression, *myc* overexpression or *cry + myc* overexpression in glial cells (Figure 4A–D’). The quantification of Myc surface signal that coincides with glial cells (anti-repo) showed that *cry* expression in glia is sufficient to increase Myc protein signal in glial cells, comparable to *myc* upregulation. In addition, *cry + myc* upregulation show a summation effect on the increase of Myc protein levels (Figure 4E). To conclude if *cry* is required for *myc* expression in GB, we quantified glial Myc signal in the control, *cry RNAi,* GB, GB + *cryRNAi* and *cry* upregulation (Figure 4F–J’). The quantifications indicate that *cryRNAi* in glial cells does not reduce the amount of Myc in glial cells (Figure 4K). In addition, GB condition triggers the number of Myc positive glial cells, as well as *cry* upregulation in glial cells (Figure 4K). Finally, *cryRNAi* expression in GB cells prevents the accumulation of Myc in GB cells. Taking all these results together, we conclude that *cry* is sufficient to trigger Myc accumulation in glial cells, and *cry* expression is necessary for Myc accumulation in GB condition.

### 2.5. Cry Contribution to GB Progression

To investigate the contribution of Cry to glioma progression, we determined the number of glial cells and volume of glial membrane network in control adult brain, GB (*PI3K + EGFR*), *PI3K + cry*, *EGFR + cry* or *myc + cry* expressed in glial cells (Figure 5A–E’). The quantification showed that all these genetic combinations cause an increase in the number of glial cells as compared to control brains (Figure 5F). However, only the GB condition provoked an expansion of the glial membrane volume, and the combination of *PI3K + cry*, *EGFR + cry* or *myc + cry* showed a volume of glial membrane comparable to control brains (Figure 5G). To further determine the contribution of *cry* to GB expansion, we analyzed the contribution of single gene upregulation in glial cells for *cry* or *myc,* and the combination of *cry + myc* expression (Figure 5H–K’). The quantification of glial cell number showed that *cry* or *myc* expression alone, or in combination, is sufficient to increase the number of glial cells with respect to control samples (Figure 5L). Nevertheless, none of these genetic modifications is sufficient to expand glial membrane volume (Figure 5M). These results suggest that *cry* or *myc* are sufficient to trigger glial cell number increase in adult brains, but not to expand the volume of glial membrane network.

### 2.6. Cry Upregulation in Glial Cells Causes Synapse Loss and Premature Death

It was previously described that GB progression induces synapse loss, an early symptom of neurodegeneration. To determine the contribution of *cry* to synapse loss, we counted the number of active zones in motor neurons of adult neuromuscular junction in the control, GB (*PI3K + EGFR*), *PI3K + cry*, *EGFR + cry* or *myc + cry* samples (Figure 5N–R). The quantification of the number of active zones showed that the expression in glial cells of GB (*PI3K + EGFR*), *PI3K + cry*, *EGFR + cry* or *myc + cry* is sufficient to reduce the number of synapses in NMJ neurons (Figure 5S).

Finally, to evaluate the systemic effect of GB and glial expression of *PI3K + cry*, *EGFR + cry* or *my c + cry*, we analyzed the lifespan of adult individuals. The results show that GB causes a premature death, as previously described in *Drosophila* and mice Xenografts [8,9,11], glial upregulation of *EGFR + cry* or *myc + cry* causes a significant reduction of lifespan but less aggressive than GB, and *PI3K + cry* upregulation in glial cells does not reduce lifespan (Figure 5T).

## 3. Discussion

Different studies have established a relation between alterations in circadian rhythm genes and cancer [32,34]. Specifically, one of the genes associated with different types of cancer is *cry* [35,36,37]. Thus, this study aims to investigate the role of *cry* in a *Drosophila* GB model.

The previous work of Luo et al. 2012 [38] describes a reduction of the number of glial cells positive for *cry1/2* expression in glioma tissue compared to normal tissue. However, the authors show that glioma cells that are positive for Cry1/2 show an increase in the amount of Cry1/2 with respect to non-tumoral tissue. Moreover, both Madden et al. 2014 [26] (with a sample 10 times larger than that of Luo et al. 2012) and Wang et al. 2021 [27] (using data from three different databases) analyzed the expression of circadian genes in glioma tissue compared to healthy tissue and conclude that *cry1* is overexpressed in glioma tissue. We also found this result in other databases such as https://www.proteinatlas.org/ and http://gliovis.bioinfo.cnio.es/, accessed on 10 January 2022, [39], which in turn is compatible with the observations in the *Drosophila* model of GB.

Nonetheless, Fan et al. [40] investigated the role of Cry2 in rat glioma cells and observed that *cry2* mRNA and protein levels showed aberrant rhythmic periodicity of 8 h, compared to 24 h in normal tissue. Thus, futures studies on the contribution of circadian rhythms genes should take into consideration the variations of expression.

On the contrary, Dong et al. [41] state that glioblastoma stem cells (GSCs) displayed robust circadian rhythms dependent on core clock transcription factors. The use of Cry1/2 agonists induced anti-tumor effects suggesting that GSCs are sensitive to *cry1/2* activity. Taken the different conclusions into consideration, most of the literature and our data suggest that *cry* is upregulated in glioma cells and promotes glioma progression; however, the role of *cry* expression in *Drosophila,* or *cry1/2* expression in mammals, may differ according to the glioma subtype, the specific mutations in glioma cells and the cell population of study within the glioma and the hour of the day.

We described an increase in *cry1 mRNA* levels in human GB samples and in a well-studied *Drosophila* model of GB. However, we cannot conclude that GB cells show an increase in *cry* transcription, or an enhancement of *cry* mRNA stability. The *Drosophila* GB model is based on the activation of the two most frequently mutated pathways in GB, PI3K and EGFR, which converge in Myc as a coincidence point. These pathways are of great relevance to promote GB cells expansion, GB progression and, in consequence, the deterioration of neighboring neurons and a premature death. The results indicate that *cry* upregulation in *Drosophila* GB cells depends on *PI3K* expression, and it is required for GB cells number increase and synapse loss (Figure 6). In addition, *cry* expression in glial cells is sufficient to increase the number of glial cells. However, *cry* expression is expendable for normal wt glial growth during development. Taking into consideration that *cry* is upregulated in GB cells and promotes glia cells number increase, we did not observe any contribution to normal glia development, which makes Cry a potential target for GB treatment.

Besides, we show that Cry is necessary and sufficient to induce *myc* expression in GB cells. This agrees with in vitro studies that revealed an increase in Myc levels as a result of *cry* upregulation [32]. Therefore, we propose that *cry* is part of the PI3K-Myc signaling pathway in GB, where *cry* upregulation would be associated with glial cells number increase. However, PI3K is a highly promiscuous enzyme that participates in numerous signaling pathways, and the results suggest that Cry contribution is restricted to the malignant features of GB dependent on *myc,* such as GB cell number increase and neurodegeneration. However, *cry* expression is independent of glial membrane expansion characteristic of GB progression. Besides, *cry* expression in glial cells partially reduces lifespan, but is less aggressive than GB. This result suggests that Cry plays a central role in GB and is required for GB formation, and *cry* mutations might be responsible for several features of GB. The human gene expression databases indicate that *cry1* expression levels correlate negatively with lifespan, and it is associated with a poor prognosis. In conclusion, these results suggest that further studies on the contribution of Cry1 to human GB progression could lead to novel strategies to treat GB patients.

Recent publications describe the communication between GB cells and neurons in human GB cells and mice xenografts based on the establishment of electrical and chemical synapses, which are essential for tumor progression [42,43]. A possible explanation for GB prevention by *cry* downregulation arises from the non-circadian function of Cry as a regulator of synapse number through the genetic and physical association with the key presynaptic protein Bruchpilot (Brp) [44,45]. In *Drosophila*, *cry* mutants show reduced *brp* expression levels. Actually, Cry interacts physically with Brp to modulate its stability, and triggers its degradation activated by light. Therefore, it is possible that *cry* overexpression in glial cells promotes the establishment of synapses with neurons. Moreover, the absence of light input impairs Brp degradation in glial cells, thus promoting tumoral progression. In conclusion, further experiments are required to unveil the molecular interactions of Cry and Brp proteins, including the putative formation of abnormal synapses between glial and neurons under the GB condition.

The studies of other groups describe the beneficial effects of haloperidol on *cry1* expression in GB cells, but these results obtained in cell culture suggest that the doses required to treat patients might be toxic; in consequence, specific delivery strategies combined with haloperidol are worth of study. In addition, we observed significant effects of *cry* knockdown in normal glial cells, in line with Bolukbasiet al., who recently described the extension of lifespan by *foxo* upregulation in glial cells [46]. We observed an effect of *cry* upregulation in the number of glial cells (Figure 5L). Given that *cry* and *foxo* respond to PI3K pathway, it is tempting to speculate that *cry* expression is relevant for lifespan extension by PI3K pathway, and associated behaviors such as diet restriction.

The classical definition of Cry as a regulator of circadian rhythms can now be expanded to the biology of glial cells, GB progression and the expansion of lifespan in *Drosophila*. This plethora of different phenotypes associated with one gene is now a common feature previously described for Troponin I [47,48,49], Caspases [50,51,52] or even other circadian genes as *per1* [53,54] and contributes to the explanation of the multiple phenotypes observed in patients.

This study describes the epistatic relationship between *PI3K*, *cry* and *myc* and the relevance for GB progression. The strengths of this study rely on the importance of understanding the mechanisms underlying the progression of a fatal tumor as GB, and the reliability of *Drosophila* as an animal model useful to study human disease. However, it is important to take into consideration the limitations of the results to put them in perspective. We used a model based on the activation of PI3K and EGFR pathways that reproduces the key features of human disease progression, but the contribution of additional mutations such as IDH or TP53 in GB require further studies; thus, new models in flies or other animal models of study will contribute to validate and narrow down our findings.

## 4. Materials and Methods

### 4.1. Fly Stocks and Genetics

All fly stocks were maintained at 25 °C (unless otherwise specified) on a 12/12 h light/dark cycles at constant humidity in a standard medium. The stocks used from Bloomington Stock Center were *tub-Gal80^TS^* (BL-7019), *Repo-Gal4* (BL-7415) and *UAS-LacZ* (BL-8529). Other fly stocks used were *UAS-dEGFR*^λ^, *UAS-dp110^CAAX^* (gift from R. Read [6]), *UAS-cry* (gift from F. Royer [55]), *GFP-cry* (BDSC_76317, gift from P.E. Hardin), *UAS-PI3K* (gift from J. Botas [56]), *UAS-cryRNAi* (gift from F. Royer [57]) and UAS-dMyc (gift from E. Moreno [58]).

The stock containing *UAS-cryRNAi* was previously generated and validated [57]; this construct produces a double-stranded RNA that corresponds to the 300–799 region of *cryRA* mRNA.

The glioma-inducing line contains the *UAS-dEGFR*^λ^ and *UAS-dp110^CAAX^* transgenes that encode for the constitutively active forms of the human orthologues PI3K and EGFR, respectively [6]. The *Repo-Gal4* line drives the *Gal4* expression to glial cells and precursors [59,60] combined with the *UAS-dEGFR*^λ^ and *UAS-dp110^CAAX^* line allow us to generate a glioma thanks to the Gal4 system [14]. To visualize glial or GB cells membrane, we induced the expression of a myristoylated form of red fluorescent protein (UAS-mRFP, described in [9]) under the control of the specific glial promoter *repo-Gal4*.

Gal80^TS^ is a repressor of the Gal4 activity at 18 °C, although at 29 °C is inactivated [61]. The *tub-Gal80^TS^* construct was used in all the crosses to avoid the lethality caused by the glioma development during the larval stage. The crosses were kept at 17 °C until the adult flies emerged. To inactivate the Gal80^TS^ protein and activate the Gal4/UAS system to allow for the expression of our genes of interest; the adult flies were maintained at 29 °C for 7 days except in the survival assay (flies were at 29 °C until death).

### 4.2. Immunostaining and Image Acquisition

All tissues were treated in simultaneously for each experiment. Adult brains were dissected and fixed with 4% formaldehyde in phosphate-buffered saline for 20 min, whereas adult NMJ were fixed for 10 min; in both cases, samples were washed 3 × 15 min with PBS + 0.4% triton, blocked for 1 h with PBS + 0.4% triton + BSA 5%, incubated overnight with primary antibodies, washed 3 × 15 min, incubated with secondary antibodies for 2 h and mounted in Vectashield mounting medium with DAPI in the case of the brains. The primary antibodies used were anti-Repo mouse (1/200; DSHB, Iowa City, IA, USA) to recognize glial nuclei, anti-Bruchpilot-nc82-mouse (1/50; DSHB, Iowa City, IA, USA) to recognize the presynaptic protein Bruchpilot, anti-HRP rabbit (1/400; Cell Signaling, Danvers, MA, USA) to recognize neuronal membranes, anti-GFP rabbit (1:500; DSHB, Iowa City, IA, USA) and anti-Myc guinea pig (1/100; DSHB, Iowa City, IA, USA) to recognize the nuclear protein Myc. The secondary antibodies used were anti-mouse, -rabbit or -guinea pig Alexa 488 or 647 (1/500; Life Technologies, Carlsbad, CA, USA). Images were taken by a Leica SP5 confocal microscopy applying same conditions for each experiment.

### 4.3. qRT-PCR

The mRNA for all samples was extracted from adult brains and processed in parallel. For this, 1- to 4-day-old male adult mice were maintained at 29 °C for 7 days and collected on dry ice at ZT6. Total RNA was extracted by triplicate from 30 heads. RNA was extracted with TRIzol and phenol chloroform. cDNA was synthetized from 1 µg of RNA and cDNA samples from 1:5 dilutions were used for real-time PCR reactions. Transcription levels were determined in a 14 µL volume in duplicate using SYBR Green (Applied Biosystems, Waltham, MA, USA) and 7500 qPCR (Thermo Fisher Scientific, Waltham, MA, USA). We analyzed transcription levels of *cry* using Rp49 as a housekeeping gene reference.

Sequences of primers were as follows.

After completing each real-time PCR run, with cycling conditions of 95 °C for 10 min, 40 cycles of 95 °C for 15 s and 55 °C for 1 min, and outlier data were analyzed using 7500 software (Applied Biosystems, Waltham, MA, USA). Ct values by triplicate of duplicates from three biological samples were analyzed calculating 2DDCt.

### 4.4. Survival Assays

Lifespan was determined under 12:12 h LD cycles at 29 °C conditions. Three replicates of 30 1- to 4-day-old male adults were collected in vials containing standard *Drosophila* media and transferred every 2–3 days to fresh *Drosophila* media.

### 4.5. Quantification

Fluorescent reporter-relative *cry* signals within brains were determined from images taken at the same confocal settings avoiding saturation. For the analysis of co-localization rates, “co-localization” tool from LAS AF Lite software (Leica, Wetzlar, Germany), was used taking the co-localization rate data for the statistics analyzing the co-localization between the green signal (both cases) and signal coming from glial tissue from three slices per brain in similar positions of the *z* axis.

Glial network was marked by a *UAS-myristoylated-RFP* reporter (mRFP) specifically expressed under the control of *repo-Gal4*. The total volume was quantified using the Imaris surface tool (Imaris 6.3.1 software, Oxford Instruments, Abingdon, UK). Glial nuclei were marked by staining with the anti-Repo (DSHB). The number of Repo + cells and number of synapses (anti-nc82; DSHB) were quantified by using the spots tool in Imaris 6.3.1 software (Oxford Instruments, Abingdon, UK). We selected a minimum size and threshold for the spot in the control samples of each experiment: 0.5 μm for active zones and 2 μm for glial cell nuclei. Myc glial signal was quantified using the Imaris surface tool (Imaris 6.3.1 software, Oxford Instruments, Abingdon, UK) creating a mask for the glial nuclei signal and exclusively selecting the myc signal corresponding to glial nuclei. Then we applied the same conditions to the analysis of the corresponding experimental sample.

### 4.6. Statistics

The results were analyzed using the GraphPad Prism 5 software. Quantitative parameters were divided into parametric and nonparametric using the D’Agostino and Pearson omnibus normality test, and the variances were analyzed with F test. The *t*-test and ANOVA test with Bonferroni’s post hoc were used in parametric parameters, using Welch’s correction when necessary. The survival assays were analyzed with Mantel–Cox test. The *p* limit value for rejecting the null hypothesis and considering the differences between cases as statistically significant was *p* < 0.05 (*). Other *p*-values are indicated as ** when *p* < 0.01 and *** when *p* < 0.001.

### 4.7. Human GB Databases

We used a public open access database (http://gliovis.bioinfo.cnio.es/, accessed on 1 February 2022) to analyze the expression of human *Cry1* gene in GB samples. We used the “Adult” samples in CGGA Dataset and included the data from primary and secondary tumor types. The data shown in Figure 1 correspond to the “expression” and “survival” tabs. Please note that nomenclature corresponds to the 2016 classification. GBM—Glioblastoma multiforme.

## Figures and Tables

**Figure 1 ijms-23-02043-f001:**
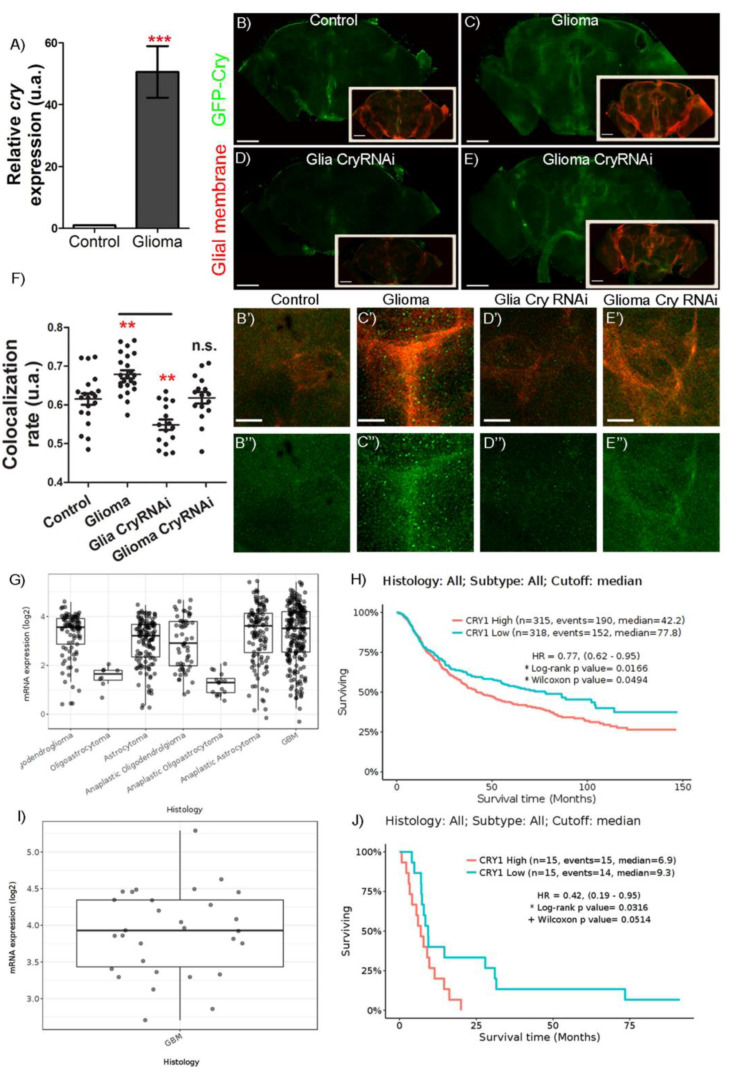
Circadian gene *cry* is upregulated in both human GB samples and GB *Drosophila* model. (**A**) RT-qPCR analysis of complete brains of 7-day-old adult flies from *repo-Gal4* > *UAS-LacZ* (Control) and *repo-Gal4* > *UAS-dEGFR**^λ^, UAS-dp110 ^CAAX^* (Glioma) genotypes in LD conditions at ZT6 for the circadian gene *cry* (*t*-test) in *n* = 90. (**B**–**E**) Confocal microscopy images of brains of 7-day-old adult flies from (**B**) *repo-Gal4 > UAS-LacZ* (Control), (**C**) *repo-Gal4 > UAS-dEGFR**^λ^, UAS-dp110 ^CAAX^* (Glioma), (**D**) *repo-Gal4 > UAS-dEGFR**^λ^, UAS-dp110 ^CAAX^, UAS-cryRNAi* (Glioma CryRNAi) and (**E**) *repo-Gal4 > UAS-cryRNAi* (Glia CryRNAi) after using (**B’**–**E″**) magnifications of the brain lobe central region, the reporter GFP-Cry in green and the glial membrane are marked in red. (**F**) Co-localization between GFP-Cry signal and the glial membrane (mRFP). Statistical analysis in at least *n* = 16 (ANOVA, post-hoc Bonferroni). (**G**) Data on overexpression of *cry1* in human primary gliomas and GB against normal tissue. (**H**) Graph showing a lower life expectancy in those patients with primary GB and *cry1* overexpressed compared to patients with primary GB with low expression of *cry1*. (**I**) Data on *cry1* expression in human secondary GB compared to normal tissue. (**J**) Graph showing a lower life expectancy in those patients with secondary GB and *cry1* overexpressed compared to patients with secondary GB with low expression of *cry1*. Images obtained from gliovis.bioinfo.cnio.es based on the 2016 classification of brain tumors (scale bar, 100 µm in (**B**–**E**) and 20 µm in (**B’**–**E’**) (n.s. not significant, ** *p*-value < 0.01, *** *p*-value < 0.001).

**Figure 2 ijms-23-02043-f002:**
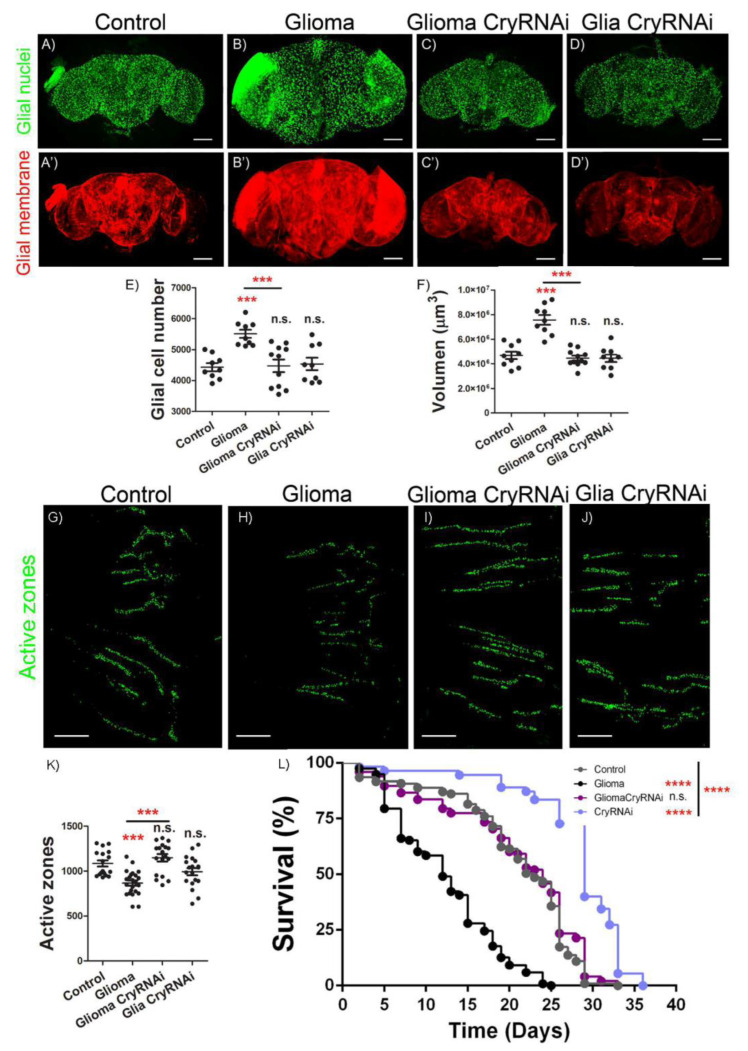
Ectopic downregulation of *cry* prevents GB tumorigenesis and effects. (**A**–**D**) Confocal microscopy images of brains from 7-day-old adult flies with the following genotypes: (**A**) *repo-Gal4 > UAS-LacZ* (Control), (**B**) *repo-Gal4 > UAS-dEGFR^λ^, UAS-dp110 ^CAAX^* (Glioma), (**C**) *repo-Gal4 > UAS-dEGFR^λ^, UAS-dp110 ^CAAX^, UAS-cryRNAi* (Glioma CryRNAi) and (**D**) *repo-Gal4 > UAS-cryRNAi* (Glia CryRNAi) with glial nuclei marked in green with anti-repo (scale bar, 100 µm). (**A’**–**D’**) Glial membrane is shown in red by the expression of mRFP. (**E**) Quantification of glial cells number and (**F**) quantification of glial membrane volume. Statistical analysis for at least *n* = 11 per genotype (ANOVA, post-hoc Bonferroni). (**G**–**J**) Confocal images of adult NMJ of 7-day-old flies from (**G**) *repo-Gal4 > UAS-LacZ* (Control) (**H**) *repo-Gal4 > UAS-dEGFR^λ^, UAS-dp110^CAAX^* (Glioma)*,* (**I**) *repo-Gal4 > UAS-dEGFR^λ^, UAS-dp110 ^CAAX^, UAS-cryRNAi* (Glioma CryRNAi) and (**J**) *repo-Gal4* > *UAS-cryRNAi* (Glia CryRNAi) genotypes. Active zones are visualized by nc82 (anti-Brp) antibody and marked in green (scale bar, 25 µm). (**K**) Quantification and statistical analysis of active zones in at least *n* = 17 per genotype (ANOVA, post-hoc Bonferroni). (**L**) Graph shows a survival assay of *repo-Gal4 > UAS-LacZ* (Control, grey), *repo-Gal4 > UAS-dEGFR^λ^, UAS-dp110^CAAX^* (Glioma, black), *repo-Gal4 > UAS-dEGFR^λ^, UAS-cryRNAi,* (Glioma *CryRNAi*, dark green) and *repo-Gal4 > UAS-cryRNAi,* (Glia *CryRNAi*, light green) flies and statistical analysis in *n* = 90 (Mantel-Cox test) (n.s. not significant, *** *p*-value < 0.001, **** *p*-value < 0.0001).

**Figure 3 ijms-23-02043-f003:**
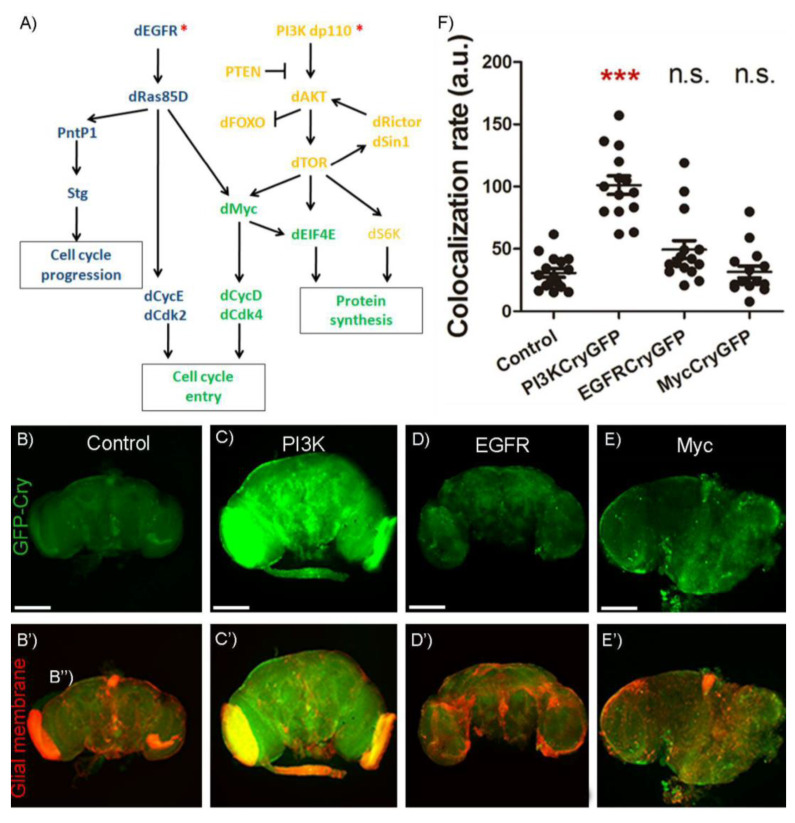
PI3K upregulates the levels of *cry*. (**A**) Scheme of EGFR (blue) and PI3K (yellow) signaling pathways involved in GB tumoral transformation with Myc as convergence point (green) (modified from [6]). (**B**–**E**) Confocal microscopy images of brains from 7-day-old adult flies with the following genotypes: (**B**) *repo-Gal4 > UAS-LacZ* (Control), (**C**) *repo-Gal4 > UAS-dp110^CAAX^* (PI3K), (**C**) *repo-Gal4 > UAS-dEGFR^λ^* (EGFR) and (**D**) *repo-Gal4 > UAS-dmyc* (Myc) using the reporter *GFP-cry* visualized in green (scale bar, 100 µm) and **B’**–**E’**) the glial membrane is marked in red by the expression of mRFP. Green and red signal merge produces the yellow signal. (**F**) Co-localization between *GFP-cry* and the glial membrane (mRFP) and statistical analysis in at least *n* = 16 (ANOVA, post-hoc Bonferroni) ( n.s. not significant, * *p*-value < 0.05, *** *p*-value < 0.001).

**Figure 4 ijms-23-02043-f004:**
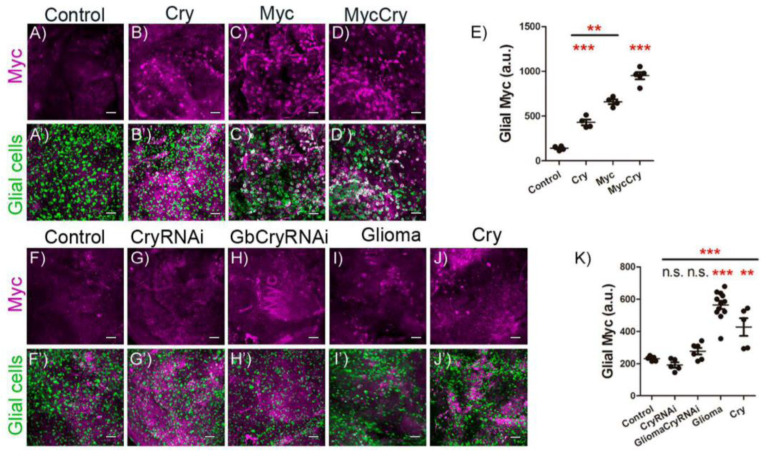
Cry increases glial Myc protein levels in physiological and GB conditions. (**A**–**D**) Confocal microscopy images of brains from 7-day-old adult flies with the following genotypes: (**A**) *repo-Gal4 > UAS-LacZ* (Control), (**B**) *repo-Gal4 > UAS-cry* (Cry), (**C**) *repo-Gal4 > UAS-dmyc* (Myc) and (**D**) *repo-Gal4 > UAS-dmyc, UAS-cry* (MycCry) with Myc marked in magenta (anti-Myc). (**A’**–**D’**) Glial nuclei marked in green (anti-Repo) (scale bar, 25 µm). (**E**) Glial Myc quantification and statistical analysis for at least *n* = 9 per genotype (ANOVA, post-hoc Bonferroni). (**F**–**J**) Confocal microscopy images of brains 7-day-old adult flies from (**F**) *repo-Gal4 > UAS-LacZ* (Control), (**G**) *repo-Gal4 > UAS-cryRNAi* (CryRNAi), (**H**) *repo-Gal4 > UAS-dEGFR^λ^, UAS-dp110 ^CAAX^, UAS-cryRNAi* (Glioma CryRNAi), (**I**) *repo-Gal4 > UAS-dEGFR^λ^, UAS-dp110 ^CAAX^* (Glioma) and (**J**) *repo-Gal4 > UAS-cry* (Cry); Myc is marked in magenta (anti-Myc) (**F’**–**J’**) and glial nuclei are marked in green (anti-Repo) (scale bar, 25 µm). (**K**) Glial Myc quantification and statistical analysis for at least *n* = 12 per genotype (ANOVA, post-hoc Bonferroni) ( n.s. not significant, ** *p*-value < 0.01, *** *p*-value <0.001).

**Figure 5 ijms-23-02043-f005:**
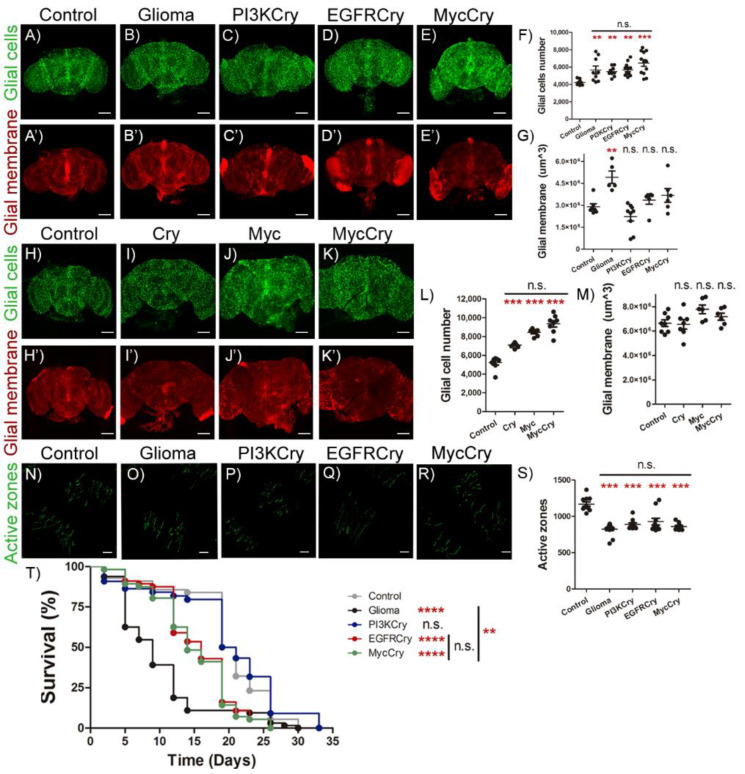
*EGFR-cry* co-expression induces glial cells number increase, synapse number and survival reduction. (**A**–**E**) Confocal microscopy images of brains from 7-day-old adult flies with the following genotypes: (**A**) *repo-Gal4 > UAS-LacZ* (Control), (**B**) *repo-Gal4 > UAS-dEGFR^λ^, UAS-dp110 ^CAAX^* (Glioma), (**C**) *repo-Gal4* > *UAS-dp110 ^CAAX^, UAS-cry* (PI3KCry), (**D**) *repo-Gal4 > UAS-dEGFR^λ^, UAS-cry* (EGFRCry) and (**E**) *repo-Gal4 > UAS-dmyc, UAS-cry* (MycCry) with glial nuclei marked in green (anti-Repo) (scale bar, 100 µm). (**A’**–**E’**) Glial membrane is visualized in red by the expression. (**F**) Glial cells number and (**G**) glial membrane volume quantification and statistical analysis for at least *n* = 12 per genotype (ANOVA, post-hoc Bonferroni). (**H**–**K**) Confocal images of adult brains of 7-day-old flies from (**H**) *repo-Gal4 > UAS-LacZ* (Control) (**I**) *repo-Gal4 > UAS-cry* (Cry)*,* (**J**) *repo-Gal4 > UAS-dmyc* (Myc) and (**K**) *repo-Gal4* > *UAS-cry, UAS-dmyc* (MycCry) genotypes with glial nuclei marked in green (anti-Repo) (scale bar, 100 µm) (**H’**–**K’**) and glial membrane shown in red (mRFP). (**L**) Glial cells number and (**M**) glial membrane volume quantification and statistical analysis for at least *n* = 9 per genotype (ANOVA, post-hoc Bonferroni). (**N**–**R**) Confocal microscopy images of NMJ of 7-day-old adult flies from (**N**) *repo-Gal4 > UAS-LacZ* (Control), (**O**) *repo-Gal4 > UAS-dEGFR^λ^, UAS-dp110 ^CAAX^* (Glioma), (**P**) *repo-Gal4 > UAS-dp110 ^CAAX^, UAS-cry* (PI3KCry), (**Q**) *repo-Gal4 > UAS-dEGFR^λ^, UAS-cry* (EGFRCry) and (**R**) *repo-Gal4 > UAS-dmyc, UAS-cry* (MycCry). Active zones are marked with anti-Brp (nc-82) visualized in green (nc82, anti-Brp) (scale bar, 25µm). (**S**) Quantification and statistical analysis of active zones in at least *n* = 13 per genotype (ANOVA, post-hoc Bonferroni). (**T**) Graph shows a survival assay of *repo-Gal4 > UAS-LacZ* (Control, grey), *repo-Gal4 > UAS-dEGFR^λ^, UAS-dp110^CAAX^* (Glioma, black), *repo-Gal4* > *UAS-dp110^CAAX^, UAS-cry* (PI3KCry, blue), *repo-Gal4 > UAS-dEGFR^λ^, UAS-cry,* (EGFRCry, red) and *repo-Gal4 > UAS-cry, UAS-dmyc* (MycCry, green) flies and statistical analysis in *n* = 90 (Mantel-Cox test) ( n.s. not significant, ** *p*-value < 0.01, *** *p*-value < 0.001, *p*-value < 0.0001).

**Figure 6 ijms-23-02043-f006:**
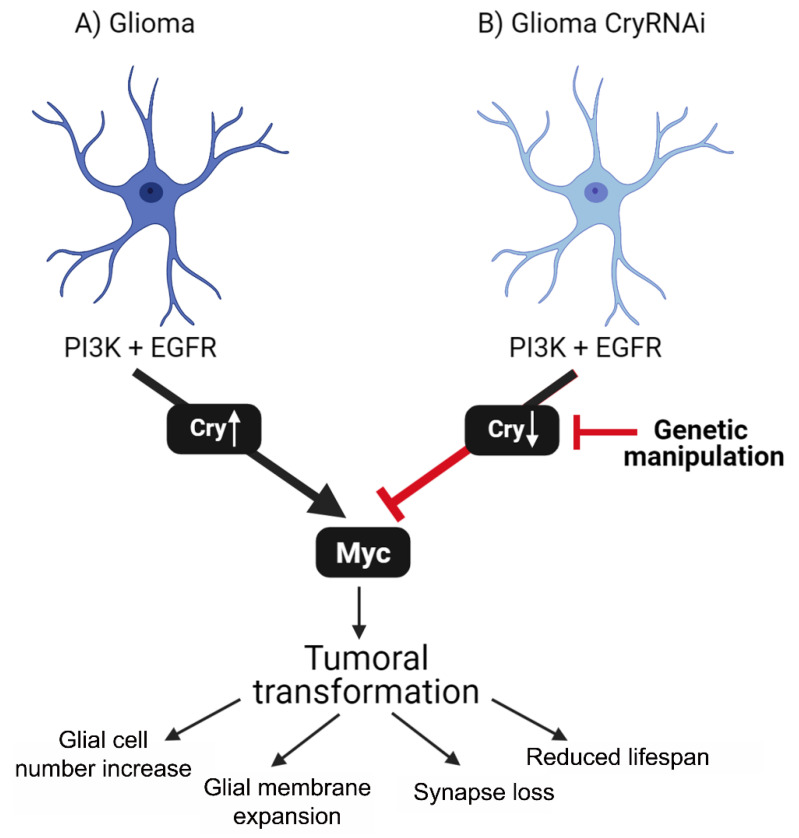
Schematic representation of Cry contribution to GB progression. (**A**) Glia cells are transformed to glioma by the co-activation of PI3K and EGFR pathways. These signals take to the accumulation of Myc in glioma cells mediated by *cry* expression. In consequence, glioma cells number increases, expand glial membrane and cause a reduction in synapse number of neurons (neurodegeneration). This process triggers premature death and reduced lifespan of the fly. (**B**) Glioma cells are transformed by PI3K and EGFR pathways activation, but upon *cry* knockdown (*cry RNAi*), they do not accumulate Myc, do not display cellular features of glioma and prevent lifespan reduction.

**Table 1 ijms-23-02043-t001:** Sequences of primers used to detect *cry* expression. *Rp49* is used as housekeeping gene.

Primer Name	5′-3′ Sequence
Rp49 F	GCATACAGGCCCAAGATCGT
Rp49 R	AACCGATGTTGGGCATCAGA
*cry* F	TTCTTCCCATCAAAACTGG
*cry* R	AAACGCATCCGATTGTAACC

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
