# Peer review of "Circadian Gene cry Controls Tumorigenesis through Modulation of Myc Accumulation in Glioblastoma Cells"

_ijms, 2022, doi:10.3390/ijms23042043_

Round 1
Reviewer 1 Report
Jarabo and colleagues presented an interesting research article aimed at elucidating the oncogenic role of the circadian gene cry in the development and progression of glioblastoma. For this purpose, the authors used a genetically-modified Drosophila GB model to study the expression of cry and how alterations in its expression may influence other oncogenes like myc. Overall, the research idea is interesting, however, the experimental design is based on the use of a single GB model. This limits the findings of the authors. Please see and address the comments below:
1) As regards the results reported in lines 109-115, the authors have to confirm these findings analyzing the TCGA GBM database;
2) The background signal of green fluorescence seems too high (Figure 1). Please try to reduce the over signal in Figure 1F;
3) All the experiments are based on the use of a single Drosophila GB model. The authors’ findings should be confirmed in GB cell lines or more complex animal models;
4) Be consistent with the use of “Gal80TS“ in chapter 4.1;
5) Provide a table containing all the information about primer sequences and thermal conditions used for the qRT-PCR experiments;
6) WB experiments for the analysis of cry protein levels are recommended.
Author Response
Rev 1
Jarabo and colleagues presented an interesting research article aimed at elucidating the oncogenic role of the circadian gene cry in the development and progression of glioblastoma. For this purpose, the authors used a genetically-modified Drosophila GB model to study the expression of cry and how alterations in its expression may influence other oncogenes like myc. Overall, the research idea is interesting, however, the experimental design is based on the use of a single GB model. This limits the findings of the authors. Please see and address the comments below:
1) As regards the results reported in lines 109-115, the authors have to confirm these findings analyzing the TCGA GBM database;
We have analyzed the data from TGCA-GBM at gliovis.bioinfo.cnio.es, and we have now included this information in the manuscript.
2) The background signal of green fluorescence seems too high (Figure 1). Please try to reduce the over signal in Figure 1F;
We have improved the background signal in Figure 1F
3) All the experiments are based on the use of a single Drosophila GB model. The authors’ findings should be confirmed in GB cell lines or more complex animal models;
We present in this manuscript the results obtained in Drosophila melanogaster GB model. This model has been validated repeatedly since the first description in 2009.
We have included in this manuscript information regarding the expression and contribution of CRY to human GB cells progression in samples from patients.
We agree with the reviewer that further investigation on the contribution of CRY in human GB samples is a need, but in this case, we consider that these experiments are out of the scope of the current manuscript.
4) Be consistent with the use of “Gal80TS“ in chapter 4.1;
We have corrected this term
5) Provide a table containing all the information about primer sequences and thermal conditions used for the qRT-PCR experiments;
This information was in the materials and methods section. We have now included a table in the Material and Methods section to facilitate the reading
RP 49 F: GCATACAGGCCCAAGATCGT
RP 49 R: AACCGATGTTGGGCATCAGA
Cry F: TTCTTCCCATCAAAACTGG
Cry R: AAACGCATCCGATTGTAACC
After completing each real-time PCR run, with cycling conditions of 95°C for 10 min, 40 cycles of 95°C for 15 seconds and 55°C for 1 min, outlier data were analyzed using 7500 software (Applied Biosystems). Ct values of triplicates from 3 biological samples were analyzed calculating 2DDCt.
6) WB experiments for the analysis of cry protein levels are recommended.
We have tried the antibody available in the field, however, we have not obtained any reliable result. Thus, we obtained a Cry-GFP line that faithfully reproduces cry expression. Unfortunately, this experiment is not possible.
Reviewer 2 Report
Concise Summary
Title: Circadian gene cry controls tumorigenesis through modulation of Myc accumulation in glioblastoma cells.
The authors aim to study the effect of cry, a regulator of circadian rhythms, on glial cells Drosophila model of glioblastoma (GB). They describe the effect of circadian gene cry in a Drosophila GB model. They study the contribution of cry to the expansion of GB cells, the neurodegeneration and to the premature death in the animal model respect with controls. In addition, it is studied different factors that regulate cry expression in this GB model, as PI3K pathway. It is concluded that cry expression contributes to the GB progression through Myc accumulation in the tumor.
Major Points
- It is stated in the article that expansion of the glial membrane or ultra-long membrane protrusions, is a relevant parameter in GB. Question: Can the authors define how the volume of glial membrane is determined?
- What is the meaning of cry expression is enough to increase the number of glial cells, but it does not expand the glial cell volume? This explanation should be incorporated to introduction.
- In the study is shown the glial cells and the deformation of the drosophila brain. However, the histology of GB is not presented. Then, how is it possible to be sure the tumor generated experimentally in drosophila is really GB?
- The authors use the parameter increase of glial cells as a relevant one in this study (lines 252-53). The use of increase of glial cells is confusing respect to the concept of tumor cell proliferation. If the authors are referring to proliferation of GB, why did the authors not use a proliferation marker to evaluate the effect of cry expression in GB proliferation? In the same sense, in the figure 5, the expression glial proliferation is used, but it does not seem be correctly used, because the authors are measuring cell density in GB, but not proliferation. Please, explain this apparent discrepancy.
- .The authors indicate that GB samples have a significant increase in the number of glial cells compared to control samples (lines 140-41). However, it is obvious that GB has more glial cells that the normal tissue. Which is the meaning of this statement?
- The authors recognize that, in Drosophila, the role of Cry expression may differ according to several factors as glioma subtype, specific mutations, tumoral cell population and the “hour of the day” (lines 315-16). Consequently, the Cry expression should be variable in GB. Then, how is it possible to state that Cry has a relevant role in GB gliomagenesis?
- The authors state that GB model in Drosophila recapitulates key aspects of the human GB both genetically and phenotypically (lines 35-36). However, the drosophila GB model could be an oversimplification of human GB. Questions: To what extent can the conclusions be extrapolated to clinical GB? Please, explain briefly the morphomolecular characteristics of the drosophila GB model respect to the human one.
- Why the authors always use the expression “glial cells” referring to cells in GB, and they never use the use the expression glial tumor cells.
- The authors state that cry upregulation in glial cells causes synapse loss. However, the synapse loss in a brain involved by GB can be produced by several parameters, as for example, nutrition or a tumor compressive effect on brain nuclei. May the authors explain why Cry expression is a relevant cause in the synapse loss?
- The authors state that Cry can now be expanded to the biology of glial cells, GB progression and life span (lines 370-71). The sentence is incomplete, confuse and too generic. The authors should say that they have showed the effects of cry in the tumor progression and life span of the drosophila affected by GB. In any case, conclusions should be reformulated in a clearer way.
Minor points
- In conclusion (lines 372-374), it is said that “is now a common feature previously described for TroponinI [47–49], Caspases [50–52] or even other circadian genes as per1 53,54] and contributes to the explanation of the multiple phenotypes observed in patients”. It mus be taken into account that in conclusions references should not be cited.
- Line 102. GFP has not been previously defined.
- Line 104. RFP has not been previously defined.
- Line 109. Glioblastoma multiforme has been previously used with acronym GB.
- Line 241. EGFP not has been defined previously.
Conclusion
In summary, I consider that it is an interesting and original work that give relevant information about the role of the cry expression in the tumorigenesis of GB. The authors conclude that Cry protein contributes to the GB progression and that it can produce neurodegeneration and reduction of life span in a drosophila model of glioma. However, there are several aspects to be modified in the text according to the reviewer comments before the manuscript could be suitable for publication.
Author Response
Rev 2
Title: Circadian gene cry controls tumorigenesis through modulation of Myc accumulation in glioblastoma cells.
The authors aim to study the effect of cry, a regulator of circadian rhythms, on glial cells Drosophila model of glioblastoma (GB). They describe the effect of circadian gene cry in a Drosophila GB model. They study the contribution of cry to the expansion of GB cells, the neurodegeneration and to the premature death in the animal model respect with controls. In addition, it is studied different factors that regulate cry expression in this GB model, as PI3K pathway. It is concluded that cry expression contributes to the GB progression through Myc accumulation in the tumor.
MajorPoints
- It is stated in the article that expansion of the glial membrane or ultra-long membrane protrusions, is a relevant parameter in GB. Question: Can the authors define how the volume of glial membrane is determined?
This is described in the Methods section: “Glial network was marked by a UAS-myristoylated-RFP reporter (mRFP) specifically expressed under the control of repo-Gal4. The total volume was quantified using Imaris surface tool (Imaris 6.3.1 software)”
- What is the meaning of cry expression is enough to increase the number of glial cells, but it does not expand the glial cell volume? This explanation should be incorporated to introduction.
The results included in this manuscript indicate that cry expression is regulated by PI3K pathway, and that Cry can regulate the accumulation of Myc in glial cells. However, cry expression does not affect glial cell membrane volume.These results are consistent with our previous report (Cell-to-cell communication mediates glioblastoma progression in Drosophila - PubMed (nih.gov)). We did not observe any significant increase of glial cell volume upon PI3K or Mycoverexpression in glial cells, suggesting that the pathway PI3K-Cry-Myc does not have a significant contribution in glial membrane expansion.
However, cry expression in glial cells produces an increase in the number of cells, as well as myc expression under the control of repoGal4. In addition, as indicated in the original publication by Read et al 2009, PLoS Genet, and in the diagram included in Figure 3A, neither PI3K pathway nor EGFR pathway alone are sufficient to reproduce all the features of GB.
In conclusion, the meaning that cry expression in glial cells does not expand glial membrane, but it does increase glial cells number, indicates that these phenotypes are consistent with the one caused by PI3K or myc expression, and therefore support the conclusions reached in this work.
- In the study is shown the glial cells and the deformation of the drosophila brain. However, the histology of GB is not presented. Then, how is it possible to be sure the tumor generated experimentally in drosophila is really GB?
This model was created by Read et al in 2009, PLoS Genet., this group did an excellent work analyzing the molecular characteristics of this GB model in Drosophila. Afterwards, various groups, including ourselves, have validated the model and obtained reproducible results in Drosophila, mice and human cells.
In particular, we would like to highlight the manuscripts published by our group that are included as references of this manuscript:
https://pubmed.ncbi.nlm.nih.gov/31846454/
https://pubmed.ncbi.nlm.nih.gov/30506943/
We describe cellular features of GB conserved in Drosophila GB model including perineuronal nests, formation of Tumor Microtubes, WNT signaling, JNK, MMPs, vesicle transport alterations, sensitivity to BFA, etc…
In conclusion, we believe that this Drosophila model of GB reproduces some of the most relevant features of human GB and it has been validated in the recent literature. However, we pay special attention in the description of the model, to make clear that this is a genetic model based on the activation of EGFR and PI3K pathways.
- The authors use the parameter increase of glial cells as a relevant one in this study (lines 252-53). The use of increase of glial cells is confusing respect to the concept of tumor cell proliferation. If the authors are referring to proliferation of GB, why did the authors not use a proliferation marker to evaluate the effect of cry expression in GB proliferation? In the same sense, in the figure 5, the expression glial proliferation is used, but it does not seem be correctly used, because the authors are measuring cell density in GB, but not proliferation. Please, explainthisapparentdiscrepancy.
We are really sorry for this error; we have now changed the legend in Figure 5. As the reviewer indicates, we should always refer to our results as an increase of glial cells number, as we do not directly use any marker of proliferation. It was shown in Read et al 2009 that glial cells in this Drosophila model enter in cell cycle, however we try to stay within the limits of the results shown in the manuscript and we like to use the correct term “increase of glial cells number” to avoid misinterpretations.
- .The authors indicate that GB samples have a significant increase in the number of glial cells compared to control samples (lines 140-41). However, it is obvious that GB has more glial cells that the normal tissue. Which is the meaning of this statement?
The full sentence is: “The results indicate that GB samples have a significant increase in the number of glial cells compared to control samples, but this increase depends on cry expression”. The meaning of this statement is that we need to monitor the number of glial cells in control and GB samples, to conclude that cry expression is necessary to cause this phenotype.
- The authors recognize that, in Drosophila, the role of Cry expression may differ according to several factors as glioma subtype, specific mutations, tumoral cell population and the “hour of the day” (lines 315-16). Consequently, the Cry expression should be variable in GB. Then, how is it possible to state that Cry has a relevant role in GB gliomagenesis?
We are not sure to fully understand this concern. The number of factors that affect GB progression, identity and prognosis is yet unknown, but includes the different mutations, and the type of cell within GB tissue. In addition, it is well known that genes, including cry, change the expression levels during the 24 hours of every day. We do not understand why the variable expression of a gene does not make it relevant for GB progression.
- The authors state that GB model in Drosophila recapitulates key aspects of the human GB both genetically and phenotypically (lines 35-36). However, the drosophila GB model could be an oversimplification of human GB. Questions: To what extent can the conclusions be extrapolated to clinical GB? Please, explain briefly the morphomolecular characteristics of the drosophila GB model respect to the human one.
We would like to thank the reviewer for this question as we are aware of the reluctances about using Drosophila as a model for human disease. Unfortunately, all animal models or cell culture have a number of limitations that are relevant for the interpretation of the results. We have been doing a great effort during more than a decade, to detect the limitations of Drosophila model of GB, and the advantages.
Still it is not clear in the field which are the most relevant features of GB for its classification, study, progression or treatment. The WHO classification of GBs has recently changed (2021) from the last description in 2016. This rapid change of classification corresponds to novel discoveries and results that still need to clarify what is the relevance of each feature of GB.
This manuscript is the first attempt to put Cry in the signaling that drives GB, but the clinical relevance of our results will have to be validated. However, so far the conclusions that we have reached using Drosophila GB model regarding WNT, JNK, MMPs, TMEM167, vesicle transport and treatment with BFA have been validated in mice Xenograft experiments. We cannot conclude that these results will be of clinical relevance until this is tested in patients, and we have not aimed to go so far in any of our conclusions.
Regarding the morphomolecular characteristics of GB, the molecular signals that mediate GB progression in humans and Drosophila were studied in detail in Read et al 2009, PLoS Genet. About the morphological characteristics of GB in flies, it is clear that the anatomy of the brain shows clear differences and therefore, this is a limitation. Thus, the studies have focused in the cellular level and Drosophila GB cells reproduce the formation of TMs, infiltrate, form perineuronal nests and most recently, we have described that form synapses with neurons (under review), as recently described for human GBs.
Please review the literature included in the introduction where various different groups have validated different characteristics of this Drosophila model, I would like to highlight the following:
https://pubmed.ncbi.nlm.nih.gov/19214224/
https://pubmed.ncbi.nlm.nih.gov/33172899/
https://pubmed.ncbi.nlm.nih.gov/33978540/
https://pubmed.ncbi.nlm.nih.gov/33526430/
https://pubmed.ncbi.nlm.nih.gov/31846454/
https://pubmed.ncbi.nlm.nih.gov/30506943/
https://pubmed.ncbi.nlm.nih.gov/30357574/
https://pubmed.ncbi.nlm.nih.gov/32457905/
- Why the authors always use the expression “glial cells” referring to cells in GB, and they never use the use the expression glial tumor cells.
We use the term GB cells when we want to indicate that these are glial tumor cells,however, in general we use glial cells as the experiments include wild type glial cells, GB samples, and GB samples with additional genetic manipulations. Be believe that this is the best way to avoid misunderstandings.
- The authors state that cry upregulation in glial cells causes synapse loss. However, the synapse loss in a brain involved by GB can be produced by several parameters, as for example, nutrition or a tumor compressive effect on brain nuclei. May the authors explain why Cry expression is a relevant cause in the synapse loss?
The results included in Figure 5 show that, overexpression of cry in glial cells is sufficient to cause a reduction of synapse number in the NMJ. All our flies are maintained under the same environmental and nutritional conditions to discard an effect of the diet, temperature, humidity or Light/Dark light cycle. Besides, dissections were done in a narrow time window (ZT2-ZT6) previously described as having the smallest variations in the synapse number. The compressive effect of the tumor is a possibility that occurs in patients, however, it is remarkable that GB patients develop further neurological symptoms after clinical resection of the tumor. Therefore, the compression could not be the only source of damage to the brain. This Drosophila model aims to reproduce the infiltrative nature of GB as GB cells are distributed throughout the brain, and they do not form a tumoral mass as a core.
The results that indicates that cry expression in glial cells is sufficient to cause a reduction in the number of synapses, is an indication that some of the features observed in GB, can be reproduced just by upregulation of cry.
- The authors state that Cry can now be expanded to the biology of glial cells, GB progression and life span (lines 370-71). The sentence is incomplete, confuse and too generic. The authors should say that they have showed the effects of cry in the tumor progression and life span of the drosophila affected by GB. In any case, conclusions should be reformulated in a clearer way.
We want to thank the reviewer for this comment, we have changed the conclusion. The extension of life span does refer to the results included in Figure 2 where cry overexpression in glial cells, significantly expands life span in flies. We have included further details in the sentence.
Minorpoints
- In conclusion (lines 372-374), it is said that “is now a common feature previously described for TroponinI [47–49], Caspases [50–52] or even other circadian genes as per1 53,54] and contributes to the explanation of the multiple phenotypes observed in patients”. It mus be taken into account that in conclusions references should not be cited.
We have corrected this
- Line 102. GFP has not been previously defined.
We have corrected this in the text
- Line 104. RFP has not been previously defined.
- We have corrected this in the text
- Line 109. Glioblastoma multiforme has been previously used with acronym GB.
Yes, we normally use the acronym GB. However, in this case, as this sentence refers to the data extracted from a database, we would like to maintain the term as it appears in the database to avoid errors.
- Line 241. EGFP not has been defined previously.
We are sorry for this error; it has been changed by the correct term EGFR.
Conclusion
In summary, I consider that it is an interesting and original work that give relevant information about the role of the cry expression in the tumorigenesis of GB. The authors conclude that Cry protein contributes to the GB progression and that it can produce neurodegeneration and reduction of life span in a drosophila model of glioma. However, there are several aspects to be modified in the text according to the reviewer comments before the manuscript could be suitable for publication.
We would like to thank the reviewer for his/her constructive comments, we have done all the effort to solve the concerns raised by this reviewer and we hope that this new version of the manuscript, is now suitable for publication.
Reviewer 3 Report
The manuscript entitled: “Circadian gene cry controls tumorigenesis through modulation of Myc accumulation in glioblastoma cells” aimed to investigate the role of deregulation of circadian rhythm genes in the glioblastoma and the underlying mechanism using a glioblastoma model of Drosophila and comparison with samples from glioblastoma patients.
The manuscript is very interesting and very useful for understanding the pathogenesis of glioblastoma that is a disease with very high mortality.
I have the following comments for the authors:
- In material and methods they have to explain how they obtained the results presented in figure 1G-J. The methodology.
- A flow chart of the study design will help the readers to understand the study design
- Add the strengths and limitations of the current study in discussion section.
Author Response
Rev 3
The manuscript entitled: “Circadian gene cry controls tumorigenesis through modulation of Myc accumulation in glioblastoma cells” aimed to investigate the role of deregulation of circadian rhythm genes in the glioblastoma and the underlying mechanism using a glioblastoma model of Drosophila and comparison with samples from glioblastoma patients.
The manuscript is very interesting and very useful for understanding the pathogenesis of glioblastoma that is a disease with very high mortality.
I have the following comments for the authors:
- In material and methods they have to explain how they obtained the results presented in figure 1G-J. The methodology.
We have included this information in the Materials and Methods section
- A flow chart of the study design will help the readers to understand the study design
We have included a new Figure 6 to facilitate the understanding of this study.
- Add the strengths and limitations of the current study in discussion section.
We have included a specific section in the discussion
Round 2
Reviewer 1 Report
The authors have improved their manuscript, however, some fundamental issues were not properly addressed. In particular, the use of a single model and the lack of confirmatory WB analyses represent critical issues that the authors have to address. Although interesting, the manuscript cannot be accepted in the present form. Resubmission is encouraged when other experiments and GB models will be added.
Reviewer 2 Report
I think this article is an interesting and comprehensive research on the effect of effect of cry, a regulator of circadian rhythms, on glial cells Drosophila model of glioblastoma The study is well designed, and the methodology is adequate to the objectives of the investigation. The results have been analyzed in a correct manner. The authors have corrected the text according to the reviewer suggestions. I think it is a nice research, and the study has been developed as an elegant scientific approach.
Reviewer 3 Report
The authors addressed all my comments. The manuscript is ready for acceptance.
This manuscript is a resubmission of an earlier submission. The following is a list of the peer review reports and author responses from that submission.
Round 1
Reviewer 1 Report
The authors present an interesting study on cry up regulation in glioma patients. Some aspects need to be considered before accepting the manuscript for publication.
-The introduction is too lengthy and reads more like a review. Please shorten the introduction to be more focused on the content of the manuscript
-The English often is a little off and should be revised over the course of the revision process.
-While the introduction is too lengthy, the discussion is too short. Please discuss your findings more critically in light of the existing literature: e.g. Luo et al 2012 who found expression in glioma cells to be lower than in glial cells. These findings are opposite to your observations.
-These observations describing a down regulation of circadian genes including cry were also published by Ye et al who found CRY to be downregulated in glioblastoma tumor tissues.
-With your findings opposing the existing literature, you need to carefully address these reservations to increase the credibility of your findings.
-Figures 4 and 5 lack scale bars
Reviewer 2 Report
The authors used a previously described Drosophila model of EGFR-Ras and PI3K-dependent human neoplastic, invasive glioma (Read et al., 2009) to determine whether signaling-induced Myc expression leads to upregulation of the circadian gene cry, whose expression is known to be inversely correlated with median survival in glioblastoma (GBM) patients. They report that cry expression is up-regulated in both human GBM and Drosophila glioma model, that down-regulation of cry hinders glioma tumorigenesis based on glial cell number and membrane volume, that PI3K but not EGFR upregulates the levels of cry, that cry increases glial Myc levels, and that EGFR-cry co-expression induces glial proliferation but reduces survival. Associated changes in neuronal synapse number suggestive of neurodegeneration are also described.
This manuscript addresses an interesting area of research about molecular circadian clocks and gliomagenesis. Unfortunately, the poor resolution of the immunofluorescence images of glial and neuronal cells (cell number, cell volume, presynaptic area) in the Drosophila glioma model on which most of the conclusions of this paper rely is simply unacceptable. Additionally, the authors did not clearly articulate their work hypothesis regarding the interactions of Myc, cry and circadian clocks in glioma, and did not clearly describe the methods used such as the fluorochrome antibody combinations to label glial and neuronal components. They also did not demonstrate a good understanding of malignant glioma tumors and did not fully review the pertinent literature.
Line 24-25: This sentence alone shows that the authors do not know what they are talking about when it comes to malignant glioma tumors. Although GBM tumor corresponds histologically to WHO grade IV, GBM is “a high-grade glioma with predominantly astrocytic differentiation, featuring… typically a diffuse growth pattern, as well as microvascular and/or necrosis, and which either lacks mutations in the IDH genes (90% of all GBM) or have a mutation in either the IDH1 (e.g., IDH1 R132H) or IDH2 gene (10% of all GBM)” (2016 WHO classification).
Line 42-50: Introduce in more detail the Drosophila glioma model while referring to the signaling pathway diagram (now Fig. 3A).
Line 112-114: Describe in more detail your work hypothesis regarding the interactions of Myc, cry and circadian clocks in glioma and how the genetic experiments will help answer these questions.
Line 130: Provide evidence that myristoylated-red fluorescent protein (mRFP) specifically labels glial cell membranes.
Line 134-135: Although cry may be transcriptionally upregulated in GBM patients, the alternative hypothesis that the stability of cry mRNA was increased was not tested.
Line 227: Do you mean “Myc surface signal”? Please explain.
Fig. 1-5. None of the immunofluorescence images are clear enough and large enough to see labeled cells or cell clusters and to reliably assess the treatment-induced changes. Compare these images with the sharp, much larger images in Read et al., 2009.
Fig. 1G. Oligoastrocytoma and anaplastic oligoastrocytoma are no longer considered to be distinct glioma entities, and molecular diagnostics tools must be used to assign these poorly defined entities to the correct diagnosis (Louis et al., 2021). Which normal tissue was used?
References
- Burchett JB, et al. MYC ran up the clock: the complex interplay between MYC and the molecular circadian clock in cancer. Int J Mol Sci. 2021 Jul 20;22(14):7761. doi: 10.3390/ijms22147761.
- Hsieh AL, et al. Misregulation of drosophila Myc disrupts circadian behavior and metabolism. Cell Rep. 2019 Nov 12;29(7):1778-1788.e4. doi: 10.1016/j.celrep.2019.10.022.
- Kelleher FC, et al. Circadian molecular clocks and cancer. Cancer Lett. 2014 Jan 1;342(1):9-18. doi: 10.1016/j.canlet.2013.09.040.
- Kozono D, et al. Dynamic epigenetic regulation of glioblastoma tumorigenicity through LSD1 modulation of MYC expression. Proc Natl Acad Sci U S A. 2015 Jul 28;112(30):E4055-64. doi: 10.1073/pnas.1501967112.
- Louis DN, et al. The 2021 WHO Classification of Tumors of the Central Nervous System: a summary. Neuro Oncol. 2021 Aug 2;23(8):1231-1251. doi: 10.1093/neuonc/noab106.
- Oktay Y, et al. IDH-mutant glioma specific association of rs55705857 located at 8q24.21 involves MYC deregulation. Sci Rep. 2016 Jun 10;6:27569. doi: 10.1038/srep27569.
- Tateishi K, et al. Myc-driven glycolysis is a therapeutic target in glioblastoma. Clin Cancer Res. 2016 Sep 1;22(17):4452-65. doi: 10.1158/1078-0432.CCR-15-2274.
- Wagner PM, et al. Adjusting the molecular clock: the importance of circadian rhythms in the development of glioblastomas and its intervention as a therapeutic strategy. Int J Mol Sci. 2021 Aug 1;22(15):8289. doi: 10.3390/ijms22158289.
Author Response
We would like to thank the reviewers for their effort to improve out manuscript, we hope that all the corrections included in this new version fulfils the requirements for publication in IJMS.
The authors used a previously described Drosophila model of EGFR-Ras and PI3K-dependent human neoplastic, invasive glioma (Read et al., 2009) to determine whether signaling-induced Myc expression leads to upregulation of the circadian gene cry, whose expression is known to be inversely correlated with median survival in glioblastoma (GBM) patients. They report that cry expression is up-regulated in both human GBM and Drosophila glioma model, that down-regulation of cry hinders glioma tumorigenesis based on glial cell number and membrane volume, that PI3K but not EGFR upregulates the levels of cry, that cry increases glial Myc levels, and that EGFR-cry co-expression induces glial proliferation but reduces survival. Associated changes in neuronal synapse number suggestive of neurodegeneration are also described.
This manuscript addresses an interesting area of research about molecular circadian clocks and gliomagenesis. Unfortunately, the poor resolution of the immunofluorescence images of glial and neuronal cells (cell number, cell volume, presynaptic area) in the Drosophila glioma model on which most of the conclusions of this paper rely is simply unacceptable.
We have changed the images to better support our conclusions and included the type of figures that were previously published in similar publications (i.e. Jarabo et al 2021 doi: 10.26508/lsa.202000693). We understand that the relevance of the images relies on the global view of the entire brain tissue, and not on a selected portion of it, therefore the images include complete brains. To facilitate the view of the representative images, we have enlarged the pictures and included magnifications to visualize further details. In addition, we have eliminated the red channel in the images of figure 2 and 5 to better visualize synapses and the differences among the four genotypes. However, all the results are quantified in the graphs and the pictures are only representative images.
Additionally, the authors did not clearly articulate their work hypothesis regarding the interactions of Myc, cry and circadian clocks in glioma, and did not clearly describe the methods used such as the fluorochrome antibody combinations to label glial and neuronal components.
We are not sure to understand this specific point regarding the hypothesis, we have focused this work in the study of Cry as a relevant player for GB progression. The upregulation of Myc levels in GB is well described in the literature (Read et al. 2009; Annibali et al. 2014). We describe the interaction of Cry with Myc in the introduction: Different studies show a relationship between Cry and Myc. C-Myc levels have been found to decrease in mice in cry1/cry2 null mutants (Liu et al. 2020). Besides, Cry expression is induced by Myc in GB cells in culture (Altman et al. 2015)
The last sentence of the introduction clarifies our hypothesis in this work: Taking into account the deregulation in the expression of circadian rhythm genes in tumor tissues in GB, and the pre-established relationship between cry and myc, which is a key player in GB, here we validate the role of Cry in the tumorigenesis and progression of GB.
We have reviewed the text, and all the fluorocromes used in each experiment are now described in Materials and in the figure legends/results section
They also did not demonstrate a good understanding of malignant glioma tumors and did not fully review the pertinent literature.
Line 24-25: This sentence alone shows that the authors do not know what they are talking about when it comes to malignant glioma tumors. Although GBM tumor corresponds histologically to WHO grade IV, GBM is “a high-grade glioma with predominantly astrocytic differentiation, featuring… typically a diffuse growth pattern, as well as microvascular and/or necrosis, and which either lacks mutations in the IDH genes (90% of all GBM) or have a mutation in either the IDH1 (e.g., IDH1 R132H) or IDH2 gene (10% of all GBM)” (2016 WHO classification).
We want to thank the reviewer for this observation, the current classification for brain tumors was updated in August 2021 and we have updated all this information in the introduction and included the new term described in the 2021 WHO classification, we have also included the corresponding reference for further discussion (Louis et al, 2021). This new classification reduces the information regarding tumor description and therefore, we have also maintained a reference of the previous 2016 classification as it was widely used, and we believe that it will facilitate the understanding for the general readers.
Line 42-50: Introduce in more detail the Drosophila glioma model while referring to the signaling pathway diagram (now Fig. 3A).
We have modified this sentence to introduce in more detail the model. In this sentence there is also a reference to the original manuscript (Read et al, 2009). Following the recommendations of other reviewer, we have shortened the introduction and therefore, we urge the reader to follow the reference to find a complete description of the model. The diagram is of special relevance as it describes the genes involved in PI3K and EGFR pathway regarding GB, we have included an explanatory sentence regarding this diagram.
Line 112-114: Describe in more detail your work hypothesis regarding the interactions of Myc, cry and circadian clocks in glioma and how the genetic experiments will help answer these questions.
This is an interesting point and we have clarified the hypothesis regarding cry and Myc. However, we do not aim to tackle any question related to the circadian clock rather than a non-circadian contribution of Cry related to PI3K pathway and Myc regulation. Regarding the contribution of circadian clock to GB progression, we have other manuscript under review where we describe the disruption of circadian rhythms in GB, and the rescue of these rhythms by genetic modulation in neurons without any intervention in GB cells.
Line 130: Provide evidence that myristoylated-red fluorescent protein (mRFP) specifically labels glial cell membranes.
We have included and explanation of the binary expression system Gal4/UAS system (Brand and Perrimon 1993) that we use to express mRFP, and induce the GB, and we have included the corresponding references. We described in 2017 (Casas-Tinto et al 2017) the specificity of the repo-Gal4 driver for glial cells used for this study, we have included a description and the reference. We have included a brief description and a reference where we validated this mRFP signal and compared with oher membrane markers (supp. Fig1 C-G´´ in Portela et al, 2019, PLoS Biology).
Line 134-135: Although cry may be transcriptionally upregulated in GBM patients, the alternative hypothesis that the stability of cry mRNA was increased was not tested.
The reviewer has brought a very interesting point about the mechanisms to enhance the total amount of protein. We cannot discard that cry mRNA stability is enhanced, but the main conclusion of the manuscript would not change. However, we have included this idea in the discussion.
Line 227: Do you mean “Myc surface signal”? Please explain.
Yes, we have corrected this in the text.
Fig. 1-5. None of the immunofluorescence images are clear enough and large enough to see labeled cells or cell clusters and to reliably assess the treatment-induced changes. Compare these images with the sharp, much larger images in Read et al., 2009.
We have now included magnifications of the images, however, we have maintained also the images of the entire brains as it is informative for the quantitative experiments included in this manuscript. The beautiful work done by Read et al. was focused in the molecular mechanisms activated in Drosophila after PI3K and EGFR pathways activation, we believe that our work presented here requires images of the entire brain.
Fig. 1G. Oligoastrocytoma and anaplastic oligoastrocytoma are no longer considered to be distinct glioma entities, and molecular diagnostics tools must be used to assign these poorly defined entities to the correct diagnosis (Louis et al., 2021). Which normal tissue was used?
We want to thank the reviewer for this comment, we are aware of this distinction and we have now simplified this figure. These data are directly extracted from the CGGA database in GlioVis, as indicated in the text. According to the description now included in the figure, the normal tissue is Non-tumor tissue.
References
- Burchett JB, et al. MYC ran up the clock: the complex interplay between MYC and the molecular circadian clock in cancer. Int J Mol Sci. 2021 Jul 20;22(14):7761. doi: 10.3390/ijms22147761.
- Hsieh AL, et al. Misregulation of drosophila Myc disrupts circadian behavior and metabolism. Cell Rep. 2019 Nov 12;29(7):1778-1788.e4. doi: 10.1016/j.celrep.2019.10.022.
- Kelleher FC, et al. Circadian molecular clocks and cancer. Cancer Lett. 2014 Jan 1;342(1):9-18. doi: 10.1016/j.canlet.2013.09.040.
- Kozono D, et al. Dynamic epigenetic regulation of glioblastoma tumorigenicity through LSD1 modulation of MYC expression. ProcNatlAcadSci U S A. 2015 Jul 28;112(30):E4055-64. doi: 10.1073/pnas.1501967112.
- Louis DN, et al. The 2021 WHO Classification of Tumors of the Central Nervous System: a summary. Neuro Oncol. 2021 Aug 2;23(8):1231-1251. doi: 10.1093/neuonc/noab106.
- Oktay Y, et al. IDH-mutant glioma specific association of rs55705857 located at 8q24.21 involves MYC deregulation. Sci Rep. 2016 Jun 10;6:27569. doi: 10.1038/srep27569.
- Tateishi K, et al. Myc-driven glycolysis is a therapeutic target in glioblastoma. Clin Cancer Res. 2016 Sep 1;22(17):4452-65. doi: 10.1158/1078-0432.CCR-15-2274.
- Wagner PM, et al. Adjusting the molecular clock: the importance of circadian rhythms in the development of glioblastomas and its intervention as a therapeutic strategy. Int J Mol Sci. 2021 Aug 1;22(15):8289. doi: 10.3390/ijms22158289.
We want to thank the reviewer for these references. We have included them in the text.
Reviewer 3 Report
*. Please explain why you choose the cry.
*. Protein level experiment should be done.
*. RNA level should be done
*. English editing should be done.
Author Response
We would like to thank the reviewers for their effort to improve out manuscript, we hope that all the corrections included in this new version fulfils the requirements for publication in IJMS.
*. Please explain why you choose the cry.
Our previous work (under review) reveals circadian behavior alterations in flies bearing the glioma model compatible with the sleep disturbances that glioblastoma patients present. A further study of this phenotype led us to analyze RNA level of different circadian genes including cry. The result shown in Fig. 1A presents a significant increase in cry mRNA levels. However, our methodology includes the use of complete brains. Therefore, to elucidate if this increase in RNA levels was due to alterations in glial cells or neurons, we used a UAS-cryRNAi under the control of the specific glial driver repo. The results showed a complete inhibition of GB phenotype and therefore, we decided to study the molecular mechanism underlying this effect.
*. Protein level experiment should be done.
Obtaining the previously described as accurate Cry antibody is difficult due to its scarcity. Agrawal et al., 2017 developed the GFP-cry line "to detect Cry with high sensitivity, we show that GFP-Cry functions similarly to endogenous Cry (Table S1; Figure 1) and confirm Cry protein expression in all brain neurons previously detected with Cry antibody (Fig 2) ".
*. RNA level should be done
We agree with the reviewer that this study would be of great importance. However, the aim of this study is to describe the relevance on Cry in GB cells, and the relation with PI3K and Myc. The molecular mechanisms that drive an increase in the amount of specific proteins is of course, of great relevance, and it should be analyzed in future experiments. Nevertheless, we believe that this is out of the scope of this manuscript, and would not contribute to the main conclusions of the manuscript.
We have only addressed this question partially and we show cry RNA levels in Fig 1A.
*. English editing should be done.
We have corrected the text with the help of a native speaker.
Reviewer 4 Report
In the present manuscript by Jarabo et al., the authors investigated the involvement of circadian gene cry in glioblastoma tumorigenesis as well as mechanism of its regulation in a Drosophila GB model.
The authors presented many interesting results and studied in detail the role of Cry in the tumorigenesis and progression of glioblastoma and the relationship between cry and myc. The research is well designed with control samples included in each experiment. The authors also searched databases to give their results an extra dimension.
I recommend this original scientific paper for the publication, with only a few minor comments.
Line 23 – please correct the sentence „Glioblastoma (GB) is the most common and aggressive type of glioma of all brain 23 tumors...“ to „Glioblastoma (GB) is the most common and aggressive tumor type among all brain tumors...“
Throughout the text check, correct and write gene names in italic (e.g. line 113 cry1/cry2)
Line 293 – please add the reference missing
Line 330 – please correct „Bolukbaso and cols“ to „Bolukbasi et al“
Line 336 - instead of „in consequence“ please write „in conclusion“
Author Response
We would like to thank the reviewers for their effort to improve out manuscript, we hope that all the corrections included in this new version fulfils the requirements for publication in IJMS.
In the present manuscript by Jarabo et al., the authors investigated the involvement of circadian gene cry in glioblastoma tumorigenesis as well as mechanism of its regulation in a Drosophila GB model.
The authors presented many interesting results and studied in detail the role of Cry in the tumorigenesis and progression of glioblastoma and the relationship between cry and myc. The research is well designed with control samples included in each experiment. The authors also searched databases to give their results an extra dimension.
I recommend this original scientific paper for the publication, with only a few minor comments.
Line 23 – please correct the sentence „Glioblastoma (GB) is the most common and aggressive type of glioma of all brain 23 tumors...“ to „Glioblastoma (GB) is the most common and aggressive tumor type among all brain tumors...“
We want to thank the reviewer for this observation. We have introduced this modification in the main text.
Throughout the text check, correct and write gene names in italic (e.g. line 113 cry1/cry2)
We have introduced these modifications in the main text.
Line 293 – please add the reference missing
We have introduced the corresponding references in the main text.
Line 330 – please correct „Bolukbaso and cols“ to „Bolukbasi et al“
We have introduced this modification in the main text.
Line 336 - instead of „in consequence“ please write „in conclusion“
We have introduced this modification in the main text.
speaker.
Round 2
Reviewer 1 Report
The authors have sufficiently addressed my concerns/reservations.
Reviewer 2 Report
I am sorry to say that the manuscript IJMS 1513833 has not been sufficiently improved to warrant publication in International Journal of Molecular Sciences. Although improvements have been made, some of the original comments were not addressed satisfactorily and even more mistakes have been uncovered in the revised manuscript. Most importantly, the RNA interference experiments throughout the paper are based on only one cry RNAi, which is a major flaw because several cry RNAi should be compared to rule out off-target effects. Furthermore, rescue experiments to determine whether cry overexpression can overcome the effects of cry RNAi are needed to demonstrate the specificity of cry effect on myc expression.
Line 24-27: The authors repeat the same inaccurate sentence “WHO grade IV diffuse oligodendroglioma and astrocytic brain tumor” and added that these types of tumors are termed “Glioblastoma, IDH-wildtype”, which is also incorrect as glioblastoma can be either IDH wildtype or IDH mutant.
Line 89-91: Although the authors’ response letter mentions that they “have clarified the hypothesis regarding cry and myc”, the corresponding text in the revised paper has not been revised and is identical to line 112-114 of the original paper.
Line 92-94: The last sentence of the introduction mentioned in the authors’ response letter should be expanded into a statement about anticipated results based upon their work hypothesis of a positive interaction between cry and myc. Terms such as “deregulation”, “pre-established relationship”, “key player” and “role” are vague and ambiguous because they do not clearly convey the simple hypothesis that increased cry expression enhances myc expression in glial cells and in glioma cells, and that this is under the regulation of the EGFR or PI3K signaling pathway.
Line 112-113: The authors should mention right here the alternative explanation of increased cry mRNA stability.
Line 343-418: The RNAi (RNA interference) experiments used throughout the manuscript are not described. Only the vectors are mentioned.
Fig. 1B: Correct the legend “…using B’-E’) the reporter GFP-Cry signal and B”-E”) the glial membrane is marked in red” because the images show the opposite colors. Which area of the adult fly brain is shown here?
Fig. 1F: What do individual data point correspond to? Individual glioma cells or fly brains?
Fig. 1G: Although the authors’ response letter mentions that this figure has been revised, it still includes oligoastrocytoma and anaplastic oligoastrocytoma. At a minimum, the authors should indicate that the figure was obtained from the TCGA database but that this classification of glioma is no longer recognized by WHO.
Fig. 1I: The term “secondary GB” refers to (grade IV) glioblastoma that originate from grade II or grade III glioma tumors. What is the meaning of “secondary GB” in this figure? Where do these data come from?
Fig. 3C’: What does the fluorescent yellow signal indicate? Is this result consistent with the hypothesized interaction between cry and myc?
Fig. 5N-5R: The active zone images cannot be seen against the black background.
Reviewer 3 Report
Reviewer's comments were not completed.